# Robotic-Assisted versus Laparoscopic Proctectomy with Ileal Pouch-Anal Anastomosis for Ulcerative Colitis: An Analysis of Clinical and Financial Outcomes from a Tertiary Referral Center

**DOI:** 10.3390/jcm11216561

**Published:** 2022-11-04

**Authors:** Jasper Max Gebhardt, Neno Werner, Andrea Stroux, Frank Förster, Ioannis Pozios, Claudia Seifarth, Christian Schineis, Carsten Kamphues, Benjamin Weixler, Katharina Beyer, Johannes Christian Lauscher

**Affiliations:** 1Department of General and Visceral Surgery, Campus Benjamin Franklin—Charité University Medicine Berlin, Hindenburgdamm 30, 12203 Berlin, Germany; 2Department of Vascular and Endovascular Surgery, University Medical Center Hamburg-Eppendorf, Martinistraße 52, 20251 Hamburg, Germany; 3Institute of Biometry and Clinical Epidemiology, Campus Mitte—Charité University Medicine Berlin, Charitéplatz 1, 10117 Berlin, Germany; 4Corporate Controlling Department, Campus Mitte—Charité University Medicine Berlin, Charitéplatz 2, 10117 Berlin, Germany; 5Department of General and Visceral Surgery, Park-Clinic Weissensee, Schönstraße 80, 13086 Berlin, Germany

**Keywords:** colorectal surgery, inpatient costs, ileal pouch-anal anastomosis, IPAA, laparoscopic surgery, restorative proctocolectomy, robotic-assisted surgery, ulcerative colitis

## Abstract

Background: Robotic-assisted colorectal surgery is gaining popularity, but limited data are available on the safety, efficacy, and cost of robotic-assisted restorative proctectomy with the construction of an ileal pouch and ileal pouch-anal anastomosis (IPAA) for ulcerative colitis (UC). Methods: A retrospective study was conducted comparing consecutively performed robotic-assisted and laparoscopic proctectomy with IPAA between 1 January 2016 and 31 September 2021. In total, 67 adult patients with medically refractory UC without proven dysplasia or carcinoma underwent surgery: 29 operated robotically and 38 laparoscopically. Results: There were no differences between both groups regarding postoperative complications within 30 days according to Clavien-Dindo classification’ grades 1–5 (51.7% vs. 42.1%, *p* = 0.468) and severe grades 3b–5 (17.2% vs. 10.5%, *p* = 0.485). Robotic-assisted surgery was associated with an increased urinary tract infection rate (*n* = 7, 24.1% vs. *n* = 1, 2.6%; *p* = 0.010) and longer operative time (346 ± 65 min vs. 281 ± 66 min; *p* < 0.0001). Surgery costs were higher when operated robotically (median EUR 10.377 [IQR EUR 4.727] vs. median EUR 6.689 [IQR EUR 3.170]; *p* < 0.0001), resulting in reduced total inpatient profits (median EUR 110 [IQR EUR 4.971] vs. median EUR 2.853 [IQR EUR 5.386]; *p* = 0.001). Conclusion: Robotic-assisted proctectomy with IPAA can be performed with comparable short-term clinical outcomes to laparoscopy but is associated with a longer duration of surgery and higher surgery costs. As experience increases, some advantages may become evident regarding operative time, postoperative recovery, and length of stay. The robotic procedure might then become cost-efficient.

## 1. Introduction

Almost 20% of patients with ulcerative colitis (UC) require surgery during their lifetime [1]. The procedure of choice is a restorative proctocolectomy with the construction of an ileal pouch with ileal pouch-anal anastomosis (IPAA), offering an effective cure when medical treatments have been insufficient while avoiding a permanent stoma. The surgical approach has evolved from an open technique to a variety of minimally invasive procedures, including the use of a robotic platform [2,3,4]. The evolution toward a less invasive approach in this young patient population was advocated early on by the clear benefits of better cosmesis [4], shorter convalescence [5], less pronounced intra-abdominal adhesions [6], and lower rates of abdominal wall herniation [7].

Despite these advantages of laparoscopy, there are also limitations, including a limited range of motion within the narrow pelvis and confined visibility when performing distal rectal dissection and subsequent anastomosis of the ileal pouch. Ongoing innovation in minimally invasive surgery has led to advances in robotic surgical technology [8,9]. The improved ergonomics, dexterity, stable visualization of the robotic platform, and equivalent safety and efficacy outcomes have contributed to its increasing use in rectal cancer surgery [10,11].

However, are these advantages relevant in benign disease when performing robotic-assisted proctectomy with IPAA for UC? The literature comparing robotic and laparoscopic proctectomy with IPAA is scarce. Most studies include only a few patients with heterogenous patient outcomes and limited data on in-hospital costs of the two approaches [12,13].

Therefore, this study aimed to present our first series of consecutive patients who underwent robotic-assisted proctectomy with IPAA for UC. We hypothesized that the robotic approach is associated with a lower 30-day postoperative complication rate, a reduced length of hospital stay (LOS), and a lower conversion rate to open surgery. We performed an in-hospital cost analysis to compare both techniques’ costs, revenues, and profits in the operating room and on the ward.

## 2. Materials and Methods

### 2.1. Patient Cohort

The study was a retrospective, comparative study performed at the Department of General and Visceral Surgery, Charité Berlin—Campus Benjamin Franklin. The department acts as a referral center for inflammatory bowel disease (IBD) and is a highly specialized center for laparoscopic colorectal surgery. Robotic-assisted colorectal resections have been regularly performed since 2016. Using our hospital’s patient database (SAP SE, Walldorf, Baden-Württemberg, Germany), we identified all adult patients (age ≥ 18 years) with UC without proven carcinoma or dysplasia who had undergone elective robotic-assisted or laparoscopic restorative proctectomy with IPAA for UC between 1 January 2016 and 31 September 2021. The 67 consecutive patients suffered from medically refractory UC and had previously undergone laparoscopic colectomy with rectal stump closure and creation of an end ileostomy as a first step according to a three-step procedure. Patients with Crohn’s disease, indeterminate colitis, or familial adenomatous polyposis were excluded. No patient was excluded beyond these criteria. The choice of the surgical procedure depended on the surgeon’s preference and the availability of the robotic platform. Patient characteristics did not influence the choice of approach. All robotic-assisted proctectomies were performed using the Da Vinci X Surgical System (Intuitive Surgical, Mountain View, Sunnyvale, CA, USA). The study was approved by the local ethical review committee (EA4/049/21) and was in accordance with the Helsinki Declaration of 1975 and its later amendments or comparable ethical standards.

Inclusion criteria:Age ≥ 18 years;Medically refractory ulcerative colitis;Elective surgery performed between 1 January 2016, and 31 September 2021;Previous laparoscopic colectomy with rectal stump closure and creation of an end ileostomy as a first step according to a three-step procedure.

Exclusion criteria:Age ≤ 18 years;Proven carcinoma or dysplasia;Crohn’s disease, indeterminate colitis, or familial adenomatous polyposis;Non-elective surgery.

### 2.2. Surgical Interventions

All five operating surgeons were experienced laparoscopic colorectal surgeons, whereas they were in their learning curve for robotic-assisted proctectomy. The surgeons completed their basic training on the robotic console in 2015, 2017, 2019, 2020, and 2021.

In robotic-assisted proctectomy, the takedown of the terminal ileostomy was conducted first, followed by placing four 8 mm robotic trocars (Intuitive Surgical, Sunnyvale, CA, USA) in a horizontal line, a few centimeters cephalad of the umbilicus. A 12 mm auxiliary trocar was placed on the right flank, and another 12 mm auxiliary trocar was inserted via a Pfannenstiel incision as an optical trocar for the conventional laparoscopic part of the operation (Applied Medical, Santa Clara, CA, USA) (Figure 1). Laparoscopic lysis of adhesions and mobilization of the mesenteric root was then performed, followed by a robotic-assisted proctectomy in the total mesorectal excision (TME) plane with the preservation of pelvic nerves. Robotic-assisted proctectomy was conducted with monopolar curved scissors (arm 1), fenestrated bipolar forceps (arm 3), and cadiere forceps (arm 4) (Intuitive Surgical, Mountain View, Sunnyvale, CA, USA;). The robotic camera was placed in arm 2 (Intuitive Surgical, Mountain View, Sunnyvale, CA, USA). The rectum was transected at the level of the pelvic floor using either a linear cutter via the Pfannenstiel incision (TA 30; Medtronic, Dublin, Ireland) or a robotic linear stapler (SureForm 45 mm green cartridge; Intuitive Surgical, Mountain View, Sunnyvale, CA, USA). If a robotic stapler was used, a 12 mm robotic arm was placed as arm 1, and stapling was conducted via this arm.

The ileal pouch was constructed extracorporeally with three staple loads (GIA 75 mm blue cartridge; Medtronic, Dublin, Ireland) with a pouch length of 12–14 cm. The type of anastomosis-hand-sewn or stapled—was at the discretion of the operating surgeon. In this cohort, without evidence of dysplasia or carcinoma, our standard anastomosis was created with circular stapling (CEEA 29 mm; Johnson & Johnson; New Brunswick, NJ, USA). Hand-sewn anastomoses were performed on rare occasions, for example, in case of technical difficulties with the stapled anastomosis. Rigid pouchoscopy with air leak test was performed regularly. In case of leakage, an endosponge was placed at the level of the anastomosis. A loop ileostomy was created in all patients.

In the laparoscopic group, proctectomy was performed laparoscopically with mesorectal excision as described previously [14,15]. The remaining steps of the operation were performed equally in both groups.

### 2.3. Outcome Measures

Preoperative, perioperative, and 30-day postoperative outcomes were retrospectively analyzed. Preoperative data included American Society of Anesthesiologists (ASA) classification, anemia, diabetes mellitus, immunosuppressive medication (corticosteroids and/ or immunomodulators) within 30 days before surgery, number of previous abdominal surgeries, and smoking status. Perioperative characteristics included the anastomotic technique (hand-sewn or stapled anastomosis), frequency of intraoperative adverse events such as damage to organ structures (lesion of the vagina, bladder, or the ureter) or serosal tears, the operative time and the conversion rate to open surgery.

The primary endpoint was the rate of 30-day postoperative complications, including any deviation of the postoperative course defined as grades 1–5 of the Clavien-Dindo classification of surgical complications [16]. Postoperative complications included anastomotic leakage (seen on abdominopelvic imaging or endoscopy), surgical site infection (SSI [defined according to Centers for Disease Control and Prevention Definition [17]), hemorrhage (defined by blood count and/or abdominopelvic imaging, leading to transfusion and/or intervention), small bowel obstruction (SBO [defined by insertion of a nasogastric tube]), urinary tract infection (UTI [defined by a positive urinary culture and symptoms]), pneumonia (defined by imaging and need of antibiotic therapy), and pulmonary artery embolism (defined by CT imaging).

Secondary endpoints were operative time, rate of conversion to open surgery, LOS, and detailed in-hospital cost analysis. Our study population’s calculated costs, revenues, and profits refer to the Department of General and Visceral Surgery, Charité Berlin—Campus Benjamin Franklin, where the study was conducted. The calculation is based on the official guidelines of the German Institute for remuneration in hospitals (InEK) for the calculation of German Diagnosis Related Groups (G-DRGs). The tool of our corporate controlling department is a graphic display that outlines the calculation of costs and the comparison of proceeds and expenses. This tool calculated the exact inpatient costs, revenues, and profits based on the case number of every single patient, including surgery costs, costs on surgical ward, medication costs, laboratory costs, and costs of diagnostic procedures. Surgery costs were mainly referred to cutting-suture time and time of anesthesia. Surgery costs also involved costs for revision surgery. Revenues were based on the G-DRGs, in which there is a fixed case-related sum regardless of the treatment of an individual patient [18]. No additional reimbursement is granted for robotic-assisted procedures in Germany. A detailed description of the composition of the underlying inpatient cost analysis can be found in previously published data of our working group [19].

### 2.4. Statistical Analysis

Parameters were depicted according to their scale and distribution with absolute and relative frequencies for categorical parameters and mean, standard deviation (SD), median, and interquartile range (IQR) for quantitative parameters. For quantitative outcomes, statistical group comparisons were performed using the t-test for independent samples. Because of the skewed distribution of some of the variables, group differences for quantitative variables were analyzed by the Mann–Whitney U test. For categorical outcomes, group comparisons were performed using cross-tabulation and the chi-square test.

Moreover, a multiple logistic regression was conducted to identify potential UTI risk factors. The analyzed independent variables were included due to a *p* value of 0.05 or less on simple regression analysis with one independent variable. Subsequently, we performed stepwise backward variable elimination with a threshold *p* > 0.1. Values of the multivariate analysis were expressed as odds ratio (OR), 95% confidence interval (CI), and *p* value. Furthermore, a multiple linear regression analysis was performed for potential risk factors of longer operative time. Again, stepwise backward variable elimination was achieved with a threshold of *p* > 0.1. Values from this second multivariate analysis were reported as regression coefficient, 95% CI, standardized coefficient, and *p* value.

*p* values of 0.05 or less were considered statistically significant. Due to the exploratory character of the analyses, no Bonferroni correction has been performed. Data analysis was performed with IBM SPSS Statistics 27 (IBM, Armonk, NY, USA).

## 3. Results

Sixty-seven patients underwent minimally invasive restorative proctectomy with IPAA, of which 29 were performed robotically and 38 laparoscopically. Slightly more patients were female (*n* = 34, 50.7%), with a mean age of 38 years (SD = 13 years) (Table 1). There was no difference in ASA classification among patients undergoing robotic-assisted or laparoscopic surgery, with the majority classified as ASA 2 (72.4% vs. 78.9%; *p* = 0.454) (Table 1). More patients in the robotic-assisted group had more than one previous abdominal surgery than in the laparoscopic group (48.3% vs. 21.1%; *p* = 0.028) (Table 1). Both groups showed no differences concerning age, sex, BMI, smoking status, diabetes, anemia, and preoperative steroid or immunomodulatory medication (Table 1).

There were no differences between both groups regarding overall postoperative complications (Clavien-Dindo classification grade 1–5) and severe postoperative complications (Clavien-Dindo grade 3b–5) (51.7% vs. 42.1%, *p* = 0.468; 17.2% vs. 10.5%, *p* = 0.485, respectively). The rate of postoperative SSI (6.9% vs. 5.3%; *p* = 1.000), postoperative ileus (6.9% vs. 13.2%; *p* = 0.459), anastomotic leakage (10.3% vs. 10.5%; *p* = 1.000), pulmonary artery embolism (3.4% vs. 0.0%; *p* = 0.433), pneumonia (3.4% vs. 0.0%; *p* = 0.433) as well as hemorrhage (3.4% vs. 7.9%; *p* = 0.628) did not differ (Table 2). Only two patients with anastomotic leakage required re-laparotomy, with both initially operated on using the robotic platform. Patients undergoing robotic-assisted proctectomy with IPAA were more likely to suffer UTI (24.1% vs. 2.6%; *p* = 0.010) (Table 2).

Multiple logistic regression analyses after backward selection revealed that robotic-assisted proctectomy with IPAA (*p* = 0.042) and ASA 2–3 (*p* = 0.031) were associated with UTI (Table 3).

Robotic-assisted surgery took, on average, longer operative time than laparoscopic surgery (346 ± 65 min vs. 281 ± 66 min; *p* < 0.0001) (Table 4). Multiple linear regression analyses after backward selection associated robotic-assisted proctectomy with IPAA (*p* < 0.0001), hand-sewn anastomosis (*p* = 0.019), and higher BMI (*p* < 0.0001) with significantly longer operative time (Table 5).

In the robotic-assisted group, one patient was converted to open surgery versus four patients in the laparoscopic group (3.4% vs. 10.5%; *p* = 0.379) (Table 4). Compared to four patients in the laparoscopic group, one patient in the robotic-assisted group showed a positive air leak test when IPAA was performed (3.4% vs. 10.5%; *p* = 0.379) (Table 4). Regarding the LOS, both groups showed no difference, with a median of 9 days (IQR 4 days) for the robotic-assisted group and a median of 8.5 days (IQR 7 days) for the laparoscopic group (*p* = 0.877) (Table 4).

The total inpatient costs did not differ between both groups (median 15.051 EUR/16.275 USD vs. median 13.243 EUR/14.320 USD; *p* = 0.118) (Table 6). The robotic-assisted approach was associated with higher surgery costs compared to the laparoscopic approach (median 10.377 EUR/11.221 USD vs. median 6.689 EUR/7.233 USD; *p* < 0.0001); total inpatient revenues did not differ (median 15.524 EUR/16.766 USD vs. median 15.524 EUR/16.766 USD; *p* = 0.256) (Table 6), resulting in reduced total inpatient profits for robotic surgery (median 110 EUR/119 USD vs. median 2.853 EUR/3.081 USD; *p* = 0.001) (Table 6). A detailed in-hospital cost analysis can be found in Table 6.

## 4. Discussion

As the safety and feasibility of robotic-assisted low anterior resection have been demonstrated, the technique has gained momentum in rectal cancer surgery [10,20]. However, robotic-assisted proctectomy with IPAA for benign disease is still rarely performed. We maintained the well-established laparoscopic steps when working outside the pelvis and used the robotic platform specifically for proctectomy, where robotic surgery is probably most beneficial. Our technique of robotic-assisted proctectomy in UC involves a TME to the level of the pelvic floor. Although not essential in benign disease, TME provides an almost bloodless plane of dissection, allowing us to identify and maintain the surgical planes during pelvic dissection.

To our knowledge, we studied the largest cohort of patients who received robotic-assisted or laparoscopic proctectomy with IPAA for UC. Our data demonstrated the safety and feasibility of robotic-assisted proctectomy with IPAA for UC. We showed equivalent 30-day postoperative complication rates for robotic-assisted and laparoscopic proctectomy with IPAA, as indicated by comparable overall (Clavien-Dindo grade 1–5) and severe complications (Clavien-Dindo grade 3b–5) (51.7% vs. 42.1% and 17.2% vs. 10.5%, respectively).

A systemic review and meta-analysis by Flynn et al. also showed no differences in 30-day postoperative complication rates between robotic-assisted and laparoscopic proctectomy with IPAA. The systematic review revealed no randomized trials, the number of patients in most studies was small, and patient cohorts were heterogeneous [12]. In contrast, our study cohort was homogeneous: all patients suffered from ulcerative colitis without evidence of dysplasia or carcinoma, were status post colectomy, and underwent proctectomy with the construction of an ileal pouch with IPAA.

In our cohort, deaths were reported in neither group. Interestingly, UTIs occurred more often in patients undergoing robotic-assisted versus laparoscopic proctectomy: 24.1% vs. 2.6%. At the same time, urinary retention was detected in neither group (defined as failure to wean off the urinary catheter after two attempts during the hospital stay with a residual urine volume ≥ 50 mL). In our study, only patients undergoing robotic-assisted proctectomy received a TME to the level of the pelvic floor, potentially injuring the hypogastric plexus and/or the pelvic splanchnic nerves/pelvic plexus, possibly resulting in higher rates of UTI and long-term urinary dysfunction. However, the study is limited by the lack of long-term data on urinary function and the short follow-up. In addition, no patient in the robotic group developed urinary retention, making pelvic nerve damage with consecutive UTI rather unlikely. In total, 17.2% of patients in the robotic group versus 7.9% of patients in the laparoscopic group had to return to the operating room postoperatively before discharge. These patients required reinsertion of a Foley catheter, which may have contributed to the higher rate of UTIs in the robotic group. Furthermore, although not significant, there were slightly more women in the robotic group than in the laparoscopic group. Since women are at higher risk for UTIs, this may have contributed to the higher UTI rate. The proportion of patients with previous abdominal surgeries was significantly higher in the robotic group. The probably higher degree of adhesions may have contributed to a more complex pelvic dissection, a longer operative time, and a consecutively higher UTI rate.

Consistent with other studies [3,21], robotic-assisted surgery took longer than laparoscopic surgery (346 min. vs. 281 min. on average). Furthermore, multiple linear regression analyses associated hand-sewn anastomosis and higher BMI with longer operative time. Interestingly, compared to others [3,8], we could not demonstrate a progressive reduction in operative time with increasing surgeon experience (data not shown) as would be expected in any new technique. The number of robotic-assisted surgeries performed by individual surgeons may have been too small to detect such a reduction.

Based on the hypothesis that the technical advantages of the robotic platform should facilitate proctectomy and avoid the need for conversion to open surgery, we compared conversion rates between both groups. Although patients in the robotic-assisted group underwent previous abdominal surgeries more often, the conversion rate was low at 3.4%, whereas the conversion rate in the laparoscopic group was 10.5%. Remarkably, the low conversion rate was achieved in the robotic group, although the number of previous abdominal surgeries was higher in the robotic group than in the laparoscopic group. However, the difference did not reach a significance level; our study cohort may have been too small to detect such a difference. In the study of Rencuzogullari, robotic-assisted proctectomy with IPAA was also not associated with a reduced conversion rate to open surgery [22]. The prospective randomized ROLARR trial revealed a decreased conversion rate for robotic-assisted versus laparoscopic low anterior resection for rectal cancer in the subgroup of male patients [20].

Consistent with other studies, we could not demonstrate shorter LOS for patients receiving robotic-assisted proctectomy with IPAA [3,21,22]. Regarding the LOS, both groups showed no difference, with a median of 9 days for the robotic-assisted group and a median of 8.5 days for the laparoscopic group.

To our knowledge, this analysis is the first detailed economic evaluation of inpatient healthcare costs comparing robotic-assisted and laparoscopic proctectomy with IPAA. Our corporate controlling team assessed healthcare costs based on the G-DRG system. The exact costs, revenues, and profits based on the case numbers of each individual patient were calculated, enabling accurate estimates. Our results suggest that robotic-assisted proctectomy with IPAA is unlikely to be cost-saving. Regarding surgery costs, the robotic-assisted approach was more expensive than the laparoscopic approach (median 10.377 EUR [11.221 USD] vs. median 6.689 EUR [7.233 USD]). The difference was mainly due to longer operative time and higher costs for robotic instruments. On the other hand, due to a comparable LOS and postoperative complication rate, surgical ward costs did not differ between the two approaches (median 4.202 EUR [4.543 USD] vs. median 3.585 EUR [3.876 USD]). The surgical costs and surgical ward costs combined constitute the total inpatient costs. These were slightly higher for robotic-assisted compared with laparoscopic proctectomy with IPAA (median 15.051 EUR [16.275 USD] vs. median 13.243 EUR [14.320 USD]), but without reaching a significance level. Robotic procedures do not create extra revenues compared to laparoscopic procedures in the G-DRG system. All these factors led to reduced total inpatient profits (median 110 EUR [IQR 4.971 EUR] vs. median 2.853 EUR [IQR 5.386 EUR]).

When considering robotic-assisted surgery, the acquisition and maintenance costs, the operational life, and the total utilization of the robotic platform per year must be considered. A detailed health economic analysis by Jayne et al. [20], based on data provided by Intuitive Surgical in 2017 (Mountain View, Sunnyvale, CA, USA), stated that “the net benefits (excluding fixed costs) of each robotic-assisted operation included in a set of cost-effective procedures must be positive, and the entire set of cost-effective procedures must have an average net benefit of at least 1.491 € (1.611 USD). On average, all robotic-assisted operations combined must exceed this figure, with each operation making at least some positive contribution.” Based on the data of our retrospective study, robotic-assisted proctectomy was associated with median higher total inpatient costs of 1.808 EUR (1.955 USD) per case. It must be considered that all surgeons were experienced laparoscopic colorectal surgeons. In contrast, they were in their learning curve for robotic-assisted proctectomy. This might have biased our results. A lower 30-day postoperative complication rate leads to reduced costs, and LOS might occur with more advanced experience regarding the robotic-assisted technique.

It seems reasonable to further develop the technology in the rational belief that the platform’s significant potential for improvement can lead to clinical benefits in the future. Robotic technology has far greater potential for further advancement than laparoscopy [23,24]. Some potential advantages of robotic proctectomy include nerve-sparing surgery with improved long-term functional outcomes and enhanced preparation to the pelvic floor with a shorter rectal stump and lower risk of cuffitis. As new competitors and further developed robotic platforms enter the market, acquisition, maintenance, and instrument costs will decrease [25]. Under these circumstances, the robotic platform will likely become a routine part of colorectal surgery, especially for complex operations such as proctectomy with IPAA.

### Limitations

This study has several limitations. First, it is a monocentric retrospective comparative study of a prospectively maintained database, limiting the analysis of postoperative outcomes. Second, we act as a referral center for IBD and are a highly specialized center for laparoscopic colorectal surgery. Thus, our patient population, perioperative settings, and operative steps may differ from outside institutions, limiting the transferability of our findings. Third, many of our patients travel long distances. Therefore, we do not have routine postoperative visits; instead, we often rely on patient phone calls and nursing communication. Thus, our 30-day postoperative complication rate may be underreported. Still, we think that the vast majority of complications were detected in our cohort due to the relatively lengthy hospital stay in Germany, our policy to initiate diagnostics early in vague suspicion of a postoperative complication, and the encouragement of patients to contact our department in case of postoperative discomfort.

Fourth, the number of patients is small but will continue to increase as our experience with robotic-assisted proctectomy grows. Due to the small number, the study might have been underpowered, thus showing no significant differences. Considering these limitations, our data suggest that robotic-assisted proctectomy is feasible and has comparable short-term outcomes to the laparoscopic approach. Fifth, patients in the robotic group had more previous abdominal surgeries. This may have biased the results regarding postoperative complications, conversion rate, hospital stay, and costs.

The study’s strength is the detailed assessment of specific clinical and financial outcomes for either robotic-assisted or laparoscopic proctectomy with IPAA. To our knowledge, this analysis is the first thorough economic evaluation of inpatient healthcare costs comparing the largest cohort of patients who received robotic-assisted or laparoscopic proctectomy with IPAA for UC. The 67 consecutive patients suffered from medically refractory UC without dysplasia or carcinoma and had previously undergone laparoscopic total abdominal colectomy as a first step according to a three-step procedure. Therefore, our cohort was homogenous and comparable; both groups were well-balanced. All robotic-assisted proctectomies were performed by surgeons already considered experts in laparoscopic colorectal surgery. They followed well-established surgical training, proctoring, and protocol using the Da Vinci X Surgical System.

## 5. Conclusions

Robotic-assisted proctectomy with IPAA is a safe and effective technique for patients with UC, showing equivalent postoperative morbidity and conversion rates compared to laparoscopic surgery. With more experience, some benefits in operative time, postoperative recovery, and long-term functional outcomes may become evident. Expenses might decrease as new competitors and further developed robotic platforms enter the market.

## Figures and Tables

**Figure 1 jcm-11-06561-f001:**
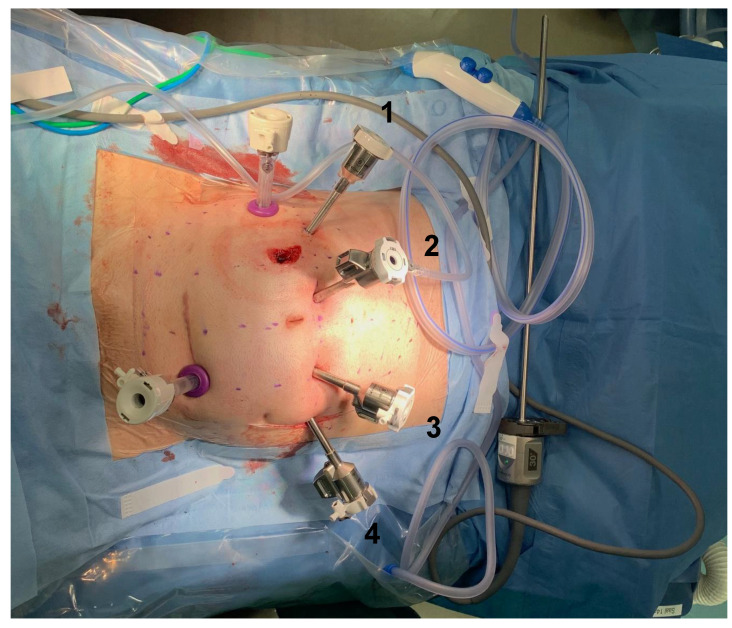
Placement of trocars after ileostomy takedown. Patient’s head, right side; patient’s legs, left side; arms 1–4, four 8 mm robotic trocars in a horizontal line, a few centimeters cephalad of the umbilicus; right flank, 12 mm auxiliary trocar; Pfannenstiel incision, 12 mm additional optical trocar for the conventional laparoscopic part of the operation.

**Table 1 jcm-11-06561-t001:** Patient Baseline Characteristics.

Variable	Robotic-Assisted Proctectomy with IPAA (*n* = 29)	Laparoscopic Proctectomy with IPAA (*n* = 38)	*p* Value
Age, Mean ± SD, years	39 ± 15	37 ± 12	0.704
ASA classification, No. (%)			0.454
1, Normal healthy patient	3 (10.3)	5 (13.2)	
2, Patient with mild systemic disease	21 (72.4)	30 (78.9)	
3, Patient with severe systemic disease	5 (17.2)	3 (7.9)	
Sex, No. (%)			0.624
Female	16 (55.2)	18 (47.4)	
Male	13 (44.8)	20 (52.6)	
BMI ^a^, Mean ± SD, kg/m^2^	24.0 ± 4.0	23.7 ± 4.6	0.699
Previous abdominal surgeries, No. (%)			**0.028 ***
1	15 (51.7)	30 (78.9)	
2–3	14 (48.3)	8 (21.1)	
Current smoking, No. (%)	2 (6.9)	0 (0.0)	0.184
Diabetes, No. (%)	1 (3.4)	1 (2.6)	1.000
Anemia, No. (%)	0 (0.0)	2 (5.3)	0.502
Preoperative steroid use, No. (%)	1 (3.4)	2 (5.3)	1.000
Preoperative immunomodulatory use, No. (%)	0 (0.0)	2 (5.3)	0.502

Data are presented as *n* (%) or mean ± SD; bold characters indicate significant values, * *p* ≤ 0.05; immunomodulatories including methotrexate, azathioprine, and biologicals. Abbreviations: ASA, American Society of Anesthesiologists; BMI, body mass index (calculated as weight in kilograms divided by height in meters squared); IPAA, ileal pouch-anal anastomosis; SD, standard deviation. ^a^ Derived from the World Health Organization classification of obesity based on BMI, with a BMI of 25.0–29.9 indicating overweight and a BMI of 30.0 or greater indicating obesity.

**Table 2 jcm-11-06561-t002:** Postoperative Complications within 30 Days.

Complications	Robotic-Assisted Proctectomy with IPAA (*n* = 29)	Laparoscopic Proctectomy with IPAA (*n* = 38)	*p* Value
Surgical complications, No. (%)			
Surgical site infection	2 (6.9)	2 (5.3)	1.000
Postoperative ileus	2 (6.9)	5 (13.2)	0.459
Anastomotic leakage	3 (10.3)	4 (10.5)	1.000
Hemorrhage	1 (3.4)	3 (7.9)	0.628
Medical complications, No. (%)			
Urinary tract infection	7 (24.1)	1 (2.6)	**0.010 ***
Pulmonary artery embolism	1 (3.4)	0 (0.0)	0.433
Pneumonia	1 (3.4)	0 (0.0)	0.433
Clavien-Dindo classification, No. (%)			0.502
Total complications, grade 1–5	15 (51.7)	16 (42.1)	0.468
Severe complications only, grade 3b–5	5 (17.2)	4 (10.5)	0.485
Return to operating room, No. (%)	5 (17.2)	3 (7.9)	0.278

Data are presented as *n* (%); bold characters indicate significant values, * *p* ≤ 0.05. Abbreviations: IPAA, ileal pouch-anal anastomosis.

**Table 3 jcm-11-06561-t003:** Multiple Logistic Regression Analysis of Potential Risk Factors for Urinary Tract Infections.

Dependent Variable	Variable	OR	95% CI		*p* Value
			Lower	Upper	
Urinary tract infection	Rob. vs. Lap.	0.097	0.010	0.922	**0.042 ***
	ASA 2–3 vs. ASA 1	7.896	1.203	51.817	**0.031 ***

The analyzed independent variables were included due to a *p* ≤ 0.05 on simple regression analysis with one independent variable. Then we performed stepwise backward variable elimination with threshold *p* > 0.1; reference categories: lap. vs. rob., robotic-assisted laparoscopic proctectomy with IPAA; ASA: ASA 2–3; data are presented as odds ratio and confidence intervals; bold characters indicate significant values, * *p* ≤ 0.05. Abbreviations: ASA, American Society of Anesthesiologists; CI, confidence interval; Lap., laparoscopic; OR, odds ratio; Rob., robotic.

**Table 4 jcm-11-06561-t004:** Perioperative Data.

Parameter	Robotic-Assisted Proctectomy with IPAA (*n* = 29)	Laparoscopic Proctectomy with IPAA (*n* = 38)	*p* Value
Operative time, Mean ± SD, min	346 ± 65	281 ± 66	**<0.0001 ***
Anastomotic technique, No (%)			1.000
Stapled ileal pouch-anal anastomosis	27 (93.1)	36 (94.7)	
Hand-sewn ileal pouch-anal anastomosis	2 (6.9)	2 (5.3)	
Conversion to open surgery, No. (%)			0.379
Yes	1 (3.4)	4 (10.5)	
No	28 (96.6)	34 (89.5)	
Intraoperative adverse event, No (%)			
None	25 (86.2)	31 (81.6)	0.745
Damage to organ structures	1 (3.4)	2 (5.3)	0.858
Positive air leak test	1 (3.4)	4 (10.5)	0.379
Serosal tear	3 (10.3)	3 (7.9)	1.000
Length of stay (days), Median (IQR)	9.0 (4.0)	8.5 (7.0)	0.877

Data are presented as *n* (%) or mean ± SD or median or interquartile range; bold characters indicate significant values, * *p* ≤ 0.05. Abbreviations: SD, standard deviation; IPAA, ileal pouch-anal anastomosis; IQR, interquartile range.

**Table 5 jcm-11-06561-t005:** Multiple Linear Regression Analysis of Potential Risk Factors for Operative Time.

Dependent Variable	Variable	Regression Coefficient	95% CI		Standardized Coefficient	*p* Value
			Lower	Upper		
Operative time	Rob. vs. Lap.	−55.953	−83.018	−28.888	−0.384	**<0.0001 ***
	Anastomotic technique	−68.393	−125.114	−11.673	−0.224	**0.019 ***
	BMI	6.491	3.367	9.616	0.385	**<0.0001 ***

The analyzed independent variables were included due to a *p* ≤ 0.05 on simple regression analysis with one independent variable. Then we performed stepwise backward variable elimination with threshold *p* > 0.1; reference categories: lap. vs. robotic, robotic-assisted proctectomy with IPAA; anastomotic technique, hand-sewn ileal pouch-anal anastomosis; data are presented as regression coefficient, confidence intervals, and standardized coefficient; bold characters indicate significant values, * *p* ≤ 0.05. Abbreviations: BMI, body mass index (calculated as weight in kilograms divided by height in meters squared); CI, confidence interval. Lap., laparoscopic, Rob., robotic.

**Table 6 jcm-11-06561-t006:** Detailed in-hospital Cost Analysis.

Category	Robotic-Assisted Proctectomy with IPAA (*n* = 29)	Laparoscopic Proctectomy with IPAA (*n* = 38)	*p* Value
Total inpatient costs			0.118
Median	15.051 EUR (16.275 USD)	13.243 EUR (14.320 USD)	
IQR	5.821 EUR (6.294 USD)	7.122 EUR (7.701 USD)	
Q25	13.547 EUR (14.649 USD)	10.907 EUR (11.794 USD)	
Q75	19.368 EUR (20.943 USD)	18.030 EUR (19.496 USD)	
Total inpatient revenues			0.256
Median	15.524 EUR (16.766 USD)	15.524 EUR (16.766 USD)	
IQR	1.544 EUR (1.668 USD)	2.129 EUR (2.299 USD)	
Q25	14.402 EUR (15.554 USD)	14.810 EUR (15.995 USD)	
Q75	15.946 EUR (17.222 USD)	16.939 EUR (18.294 USD)	
Total inpatient profits			**0.001 ***
Median	110 EUR (119 USD)	2.853 EUR (3.081 USD)	
IQR	4.971 EUR (5.369 USD)	5.386 EUR (5.817 USD)	
Q25	−3.691 EUR (−3.986 USD)	526 EUR (568 USD)	
Q75	1.280 EUR (1.382 USD)	5.911 EUR (6.384 USD)	
Surgery costs			**<0.0001 ***
Median	10.377 EUR (11.221 USD)	6.689 EUR (7.233 USD)	
IQR	4.727 EUR (5.111 USD)	3.170 EUR (3.427 USD)	
Q25	9.121 EUR (9.862 USD)	5.281 EUR (5.710 USD)	
Q75	13.849 EUR (14.975 USD)	8.452 EUR (9.139 USD)	
Surgery revenues			0.604
Median	6.970 EUR (7.528 USD)	6.671 EUR (7.205 USD)	
IQR	1.065 EUR (1.150 USD)	511 EUR (552 USD)	
Q25	6.276 EUR (6.778 USD)	6.459 EUR (6.976 USD)	
Q75	7.341 EUR (7.928 USD)	6.970 EUR (7.528 USD)	
Surgery profits			**<0.0001 ***
Median	−4.134 EUR (−4.465 USD)	−78 EUR (−84 USD)	
IQR	4.097 EUR (4.425 USD)	2.666 EUR (2.879 USD)	
Q25	−6.786 EUR (−7.329 USD)	−1.482 EUR (−1.601 USD)	
Q75	−2.689 EUR (−2.904 USD)	1.183 EUR (1.278 USD)	
Surgical ward costs			0.430
Median	4.202 EUR (4.543 USD)	3.585 EUR (3.876 USD)	
IQR	1.966 EUR (2.125 USD)	3.598 EUR (3.890 USD)	
Q25	3.603 EUR (3.896 USD)	2.858 EUR (3.090 USD)	
Q75	5.570 EUR (6.023 USD)	6.456 EUR (6.981 USD)	
Surgical ward revenues			0.305
Median	7.085 EUR (7.652 USD)	7.085 EUR (7.652 USD)	
IQR	1.126 EUR (1.216 USD)	770 EUR (832 USD)	
Q25	6.116 EUR (6.605 USD)	6.469 EUR (6.987 USD)	
Q75	7.242 EUR (7.821 USD)	7.239 EUR (7.818 USD)	
Surgical ward profits			0.957
Median	2.467 EUR (2.664 USD)	2.506 EUR (2.706 USD)	
IQR	1.945 EUR (2.101 USD)	2.718 EUR (2.935 USD)	
Q25	1.400 EUR (1.512 USD)	782 EUR (845 USD)	
Q75	3.345 EUR (3.613 USD)	3.500 EUR (3.780 USD)	

Data are EUR and USD; bold characters indicate significant values, * *p* ≤ 0.05; exchange rate from EUR in USD based on 04/15/2022. Abbreviations: IPAA, ileal pouch-anal anastomosis; IQR, interquartile range; Q, quartile, USD, United States Dollar.

## Data Availability

Publicly available datasets were analyzed in this study. These data can be found here: https://www.synapse.org/#!Synapse:syn38741915; accessed on 2 October 2022.

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
