# Peer review of "Robotic-Assisted versus Laparoscopic Proctectomy with Ileal Pouch-Anal Anastomosis for Ulcerative Colitis: An Analysis of Clinical and Financial Outcomes from a Tertiary Referral Center"

_jcm, 2022, doi:10.3390/jcm11216561_

Round 1

Reviewer 1 Report

Dear authors

This article sounds very good for the interest of robotic thoracic surgery in complex surgical procedures like the one analysed in this study. The results seem optimal for this type of surgery, especially for a team of surgeons analysing their results during their early learning curve in robotic rectal surgery. 

My major concerns are on the following points that should be clarified in the study.

The first is the higher incidence of urinary tract infection complications in the robotic surgery group. How do you explain this? 

Secondly, how do you explain that the intraoperative cost being higher in the robotic group, there is no difference in total costs? Hospital stay and postoperative complications were similar so the lack of significant difference is not well understood. Could it also be attributable to an insufficient patient sample to find a difference?

Third, there is a significant difference in both groups in the rate of patients with 1 previous major abdominal surgery in the laparoscopic group and 2-3 previous surgeries in the robotic group; as you know very well, this could be a bias in patient selection due to lack of homogeneity, given that the presence of these previous surgeries increases the duration of a procedure, possible surgical complications and possibly the rate of need for re-conversion. In any case, this should be stated in the limitations section.

We look forward to hearing from you regarding these concerns.

Kind regards

Author Response

First, we would like to thank the reviewer for his detailed comments, which helped us to improve our manuscript substantially. According to the reviewers suggestions, we have revised the manuscript and highlighted all changes in red in the revised manuscript.

Below, we address the reviewers concerns point by point (reviewers comments in italics).

This article sounds very good for the interest of robotic thoracic surgery in complex surgical procedures like the one analysed in this study. The results seem optimal for this type of surgery, especially for a team of surgeons analysing their results during their early learning curve in robotic rectal surgery. My major concerns are on the following points that should be clarified in the study.

The first is the higher incidence of urinary tract infection complications in the robotic surgery group. How do you explain this?

Thank you for pointing out this essential aspect. We believe that the relatively high urinary tract infection (UTI) rate in robotic-assisted proctectomy with ileal pouch-anal anastomosis (IPAA) might be influenced by several aspects:

  • Without indicating a significance level, 17.2 % of patients (n = 5) operated on robotically compared with 7.9 % (n = 3) operated on laparoscopically had to return to the operating room for postoperative complications. They received another Foley catheter perioperatively, possibly leading to UTIs (see page 10, lines 341 – 344). Two robotic cases were found to have UTIs after reoperation, whereas none were found in the laparoscopic group.
  • Women are more frequently affected by UTIs than men. In our study, slightly more women than men (without reaching significance level) were operated on robotically, possibly affecting UTI rate for robotic-assisted proctectomy with IPAA (see page 10, lines 344 – 347).
  • In addition, more patients in the robotic group (n = 14, 48.3%) than in the laparoscopic group (n = 8, 21.1%) had previous abdominal surgeries (p = 0.028). This may have led to adhesions, which increased operative time and resulted in a more complex pelvic dissection, which may have contributed to the higher UTI rate in the robotic group (page 10, lines 347 – 350).
  • UTIs are considered the most common bacterial infections. In a comparatively small patient cohort of 29 patients who underwent robotic-assisted proctectomy with IPAA, our finding could be a statistical coincidence and not directly related to the type of procedure. This result needs to be clarified in a larger cohort of patients.

In our study, only patients who underwent robotic-assisted proctectomy received a TME to the level of the pelvic floor with a potentially higher risk of pelvic plexus injury compared to laparoscopic proctectomy performed transmesorectally. Still, no patient in the robotic group had relevant urinary retention and left the hospital with a Foley catheter inserted, making pelvic nerve injury with consecutive UTI rather unlikely in the robotic group (see page 10, lines 335 – 341).

Secondly, how do you explain that the intraoperative cost being higher in the robotic group, there is no difference in total costs? Hospital stay and postoperative complications were similar so the lack of significant difference is not well understood. Could it also be attributable to an insufficient patient sample to find a difference?

Thank you for this important comment. The significant difference in surgical costs (median 10.377 € [11.221 USD] vs. median 6.689 € [7.233 USD]; p < 0.0001*) was mainly due to longer operative time and higher costs for robotic instruments (see page 11, lines 380 – 383). On the other hand, due to a comparable length of stay and postoperative complication rate, surgical ward costs did not differ between the two procedures (median 4.202 € [4.543 USD] vs. median 3.585 € [3.876 USD]; p = 0.430) (see page 11, lines 383 – 385). The surgical costs and surgical ward costs combined constitute the total inpatient costs. The latter were slightly higher for robotic-assisted compared with laparoscopic proctectomy with IPAA (median 15.051 € [16.275 USD] vs. median 13.243 € [14.320 USD]), but without reaching a significance level (p = 0.118) (see page 11, lines 387 – 389). One cannot tell with certainty if a significant difference in total costs will be detected in a bigger patient cohort.

Third, there is a significant difference in both groups in the rate of patients with one previous major abdominal surgery in the laparoscopic group and 2-3 previous surgeries in the robotic group; as you know very well, this could be a bias in patient selection due to lack of homogeneity, given that the presence of these previous surgeries increases the duration of a procedure, possible surgical complications and possibly the rate of need for re-conversion. In any case, this should be stated in the limitations section.

Thank you very much for this valuable input. This could indeed be a bias in patient selection. As requested, we discuss this aspect in the limitations section (see page 12, lines 435 – 437). The significant difference of 2-3 previous surgeries between the robotic-assisted and laparoscopic approaches (p = 0.028) may have influenced operative time. A highly relevant association with operative time seems unlikely since we included previous surgeries as a variable in the multiple linear regression analysis of potential risk factors for operative time and could not demonstrate a significant association. However, the number of previous surgeries might have influenced postoperative complications such as UTIs, conversion rate, surgery costs, and total inpatient costs. We have added this important information in several parts of the discussion section (page 10, lines 348 – 350; pages 11, lines 362 – 364).

We hope that this new revision of our manuscript adds sufficient further information to allow publication in the special issue “Surgery for Inflammatory Bowel Disease: State of the Art and Future Perspectives” by the Journal of Clinical Medicine.

Yours sincerely,

Jasper M. Gebhardt and Johannes C. Lauscher

Reviewer 2 Report

The authors should be congratulated for the excellent manuscript evaluating the outcomes of consecutive UC minimally invasive cases. The manuscript is clear, concise, and well-written. The methodology and statistical analysis are precise and sound. The results are shown correctly and clearly to the reader. The cost analysis is very interesting and shows the first serious analysis of the economic impact of both laparoscopic and robotic techniques for UC. The authors describe in detail the limitations and strengths of the paper. I believe future studies from the same department will show even better data when all the surgical team will acquire full expertise with the robotic approach. 

Author Response

First, we would like to thank the reviewers for their detailed comments, which helped us to improve our manuscript substantially. According to the reviewers' suggestions, we have revised the manuscript and highlighted all changes in red in the revised manuscript.

Below, we address the reviewer's concerns point by point (reviewers' comments in italics).

The authors should be congratulated for the excellent manuscript evaluating the outcomes of consecutive UC minimally invasive cases. The manuscript is clear, concise, and well-written. The methodology and statistical analysis are precise and sound. The results are shown correctly and clearly to the reader. The cost analysis is very interesting and shows the first serious analysis of the economic impact of both laparoscopic and robotic techniques for UC. The authors describe in detail the limitations and strengths of the paper. I believe future studies from the same department will show even better data when all the surgical team will acquire full expertise with the robotic approach. 

We thank the reviewer for this very appreciative assessment of our work. We hope to make further valuable contributions to robotic colorectal surgery in the future.

We hope that this new revision of our manuscript adds sufficient further information to allow publication in the special issue “Surgery for Inflammatory Bowel Disease: State of the Art and Future Perspectives” by the Journal of Clinical Medicine.

Yours sincerely,

Jasper M. Gebhardt and Johannes C. Lauscher

Reviewer 3 Report

There are many studies on topic , even systematic reviews,what is new in your study?

Introduction is too long.

 Patients were allocated into groups, randomly or  consecutively need to be made more clear ?

Inclusion exclusion criteria needs to more clear and ideally in bullet form

Could you present logic of measuring outcome of urinary tract infection (UTI [defined by a positive urinary culture and symptoms])and  pulmonary artery  embolism , these are general complications ? All TME has predisposition.

Author Response

First, we would like to thank the reviewer for his detailed comments, which helped us to improve our manuscript substantially. According to the reviewer's suggestions, we have revised the manuscript and highlighted all changes in red in the revised manuscript.

Below, we address the reviewer's concerns point by point (reviewers' comments in italics).

There are many studies on topic, even systematic reviews, what is new in your study?

Thank you for this important comment. Indeed, there are studies on this topic. Still, to the best of our knowledge, this analysis is the first detailed economic evaluation of inpatient healthcare costs with exact individual inpatient costs / profits, surgery costs / profits, and surgery ward costs / profits, comparing robotic-assisted and laparoscopic proctectomy with IPAA. We demonstrated that the robotic-assisted approach was associated with higher surgery costs compared with the laparoscopic approach (median 10.377 € [11.221 USD] vs. median 6.689 € [7.233 USD]; p  < 0.0001), while surgical ward costs (median 4.202 € [4.543 USD] vs. median 3.585 € [3.876 USD]; p = 0.430) and total inpatient revenues (median 15.524 € [16.766 USD] vs. median 15.524  € [16.766 USD]; p = 0.256) did not differ, resulting in lower total inpatient profits for robotic-assisted surgery (median 110 € [IQR 4.971 €] vs. median 2.853 € [IQR 5.386 €]; p < 0.0001) (see page 11, lines 380 – 391)

To the best of our knowledge, we have studied the largest cohort of patients who received robotic-assisted or laparoscopic proctectomy with IPAA for UC. The 67 consecutive patients suffered from medically refractory UC without dysplasia or carcinoma and had previously undergone laparoscopic total abdominal colectomy as a first step according to a three-step procedure. Therefore, our cohort was homogenous and comparable; both groups were well-balanced (see page 12, lines 439 – 448). All patients received IPAA, and the surgical procedure was well-standardized and described in detail. All robotic-assisted proctectomies were performed by surgeons already considered experts in laparoscopic colorectal surgery. They followed well-established surgical training, proctoring, and protocol using the Da Vinci X Surgical System (see pages 3 – 4, paragraph “2.2. Surgical intervention”, lines 106 – 145).

Another strength of the study lies in the detailed evaluation of the specific clinical outcomes of robotic-assisted or laparoscopic proctectomy with IPAA. We report a detailed analysis of postoperative complications within 30 days, including the Clavien-Dindo classification, highlighting severe grades 3b-5 (see table 2).

Introduction is too long.

Thank you for alerting us on this topic. We again carefully read the introduction and shortened it.

Patients were allocated into groups, randomly or consecutively need to be made more clear?

Thank you for this valuable remark. There was no randomization. All Patients were included consecutively according to the inclusion and exclusion criteria (see pages 2 – 3, paragraph “2.1. Patient cohort”, lines 93 – 104). No patient was excluded beyond these criteria. The choice of surgical procedure depended on the surgeon's preference and the availability of the robotic platform. Patient characteristics did not influence this choice. We have clarified these informations in the manuscript (see page 2, lines 85 – 87).

Inclusion exclusion criteria needs to more clear and ideally in bullet form.

Thank you for this important comment. For clarification, we list the inclusion and exclusion criteria in the method section in bullet form (see pages 2 – 3, paragraph “2.1. Patient cohort”, lines 93 – 104).

Could you present logic of measuring outcome of urinary tract infection (UTI [defined by a positive urinary culture and symptoms]) and pulmonary artery embolism, these are general complications? All TME has predisposition?

We measured all postoperative complications –surgical complications such as SSI and surgery-related complications such as pneumonia or urinary tract infections – according to the well-established classification of Clavien and Dindo. In addition, we analyzed each complication separately and compared patients undergoing robotic-assisted versus laparoscopic proctectomy with IPAA. We found a difference in UTI rates between the robotic-assisted and laparoscopic groups. To investigate this further, we performed a multivariate analysis. We do not have a clear explanation for why UTIs were more frequent in the robotic group, but we believe that several aspects may influence UTI rates:

  • Without indicating a significance level, 17.2 % of patients (n = 5) operated on robotically compared with 7.9 % (n = 3) who operated laparoscopically had to return to the operating room for postoperative complications. They received another Foley catheter perioperatively, possibly leading to UTIs (see page 10, lines 341 – 344). Two robotic cases were found to have UTIs after reoperation, whereas none were found in the laparoscopic group.
  • Women are more frequently affected by UTIs than men. In our study, slightly more women than men (without reaching significance level) were operated on robotically, possibly affecting UTI rate for robotic-assisted proctectomy with IPAA (see page 10, lines 344 – 347).
  • In addition, more patients in the robotic group (n = 14, 48.3%) than in the laparoscopic group (n = 8, 21.1%) had previous abdominal surgeries (p = 0.028). This may have led to adhesions, which increased operative time and resulted in a more complex pelvic dissection, which may have contributed to the higher UTI rate in the robotic group (page 10, lines 347 – 350).
  • UTIs are considered the most common bacterial infections. In a comparatively small patient cohort of 29 patients who underwent robotic-assisted proctectomy with IPAA, our finding could be a statistical coincidence and not directly related to the type of procedure. This result needs to be clarified in a larger cohort of patients.

In our study, only patients who underwent robotic-assisted proctectomy received a TME to the level of the pelvic floor with a potentially higher risk of pelvic plexus injury compared with laparoscopic proctectomy performed transmesorectally. Still, no patient in the robotic group had relevant urinary retention and left the hospital with a Foley catheter inserted, making pelvic nerve injury with consecutive UTI rather unlikely in the robotic group (see page 10, lines 335 – 341). Taken together, we do not think our data shows that robotic TME influences the UTI rate.

We hope that this new revision of our manuscript adds sufficient further information to allow publication in the special issue “Surgery for Inflammatory Bowel Disease: State of the Art and Future Perspectives” by the Journal of Clinical Medicine.

Yours sincerely,

Jasper M. Gebhardt and Johannes C. Lauscher